# Influence of Retrogression Time on the Fatigue Crack Growth Behavior of a Modified AA7475 Aluminum Alloy

**DOI:** 10.3390/ma16072733

**Published:** 2023-03-29

**Authors:** Xu Zheng, Yi Yang, Jianguo Tang, Baoshuai Han, Yanjin Xu, Yuansong Zeng, Yong Zhang

**Affiliations:** 1School of Materials Science and Engineering, Central South University, Changsha 410083, China; 2ALG Aluminium Inc., Guangxi Key Laboratory of Materials and Processes of Aluminum Alloys, Nanning 530031, China; 3Shanxi Aircraft Industry Corporation, Ltd., Hanzhong 723213, China; 4Key Laboratory of Non-Ferrous Metals Science and Engineering, Ministry of Education, Changsha 410083, China; 5AVIC Manufacturing Technology Institute, Beijing 100024, China

**Keywords:** retrogression and re-ageing, Al-Zn-Mg alloys, fatigue crack growth, η’ precipitates, AA7475 aluminum alloy

## Abstract

This paper investigates the effect of retrogression time on the fatigue crack growth of a modified AA7475 aluminum alloy. Tests including tensile strength, fracture toughness, and fatigue limits were performed to understand the changes in properties with different retrogression procedures at 180 °C. The microstructure was characterized using scanning electron microscopy (SEM) and transmission electron microscopy (TEM). The findings indicated that as the retrogression time increased, the yield strength decreased from 508 MPa to 461 MPa, whereas the fracture toughness increased from 48 MPa√m to 63.5 MPa√m. The highest fracture toughness of 63.5 MPa√m was seen after 5 h of retrogression. The measured diameter of η’ precipitates increased from 6.13 nm at the retrogression 1 h condition to 6.50 nm at the retrogression 5 h condition. Prolonged retrogression also increased the chance of crack initiation, with slower crack growth rate in the long transverse direction compared to the longitudinal direction. An empirical relationship was established between fracture toughness and the volume fraction of age-hardening precipitates, with increasing number density of precipitates seen with increasing retrogression time.

## 1. Introduction

Aluminum alloys made of high-strength Al-Zn-Mg-Cu have been a popular choice for use in aerospace and other civilian applications since the 1940s [1]. These alloys achieve their highest strength through peak aging conditions; however, they are also susceptible to stress corrosion cracking (SCC). Moreover, the safe life design approach necessitates a high level of fracture toughness and a low fracture cracking growth rate. Therefore, the fatigue crack growth behavior under spectrum loading is increasingly being considered when choosing alloys for aircraft structures that are critical to fatigue.

The AA7475 aluminum alloy is a good combination of high strength and superior fatigue crack growth resistance when compared with AA7050 and AA7075 in comparable tempers [2,3,4]. The AA7475 alloy originates from AA7075 and AA7175 aluminum alloys. It has a higher zinc to magnesium ratio, meaning a greater zinc content and less magnesium content. The copper content range is comparable between the two alloys, with AA7075 having 1.2–2.0% weight compared to 1.2–1.9% weight in AA7475. AA7475 and AA7075 alloys are both categorized as Al-Zn-Mg-Cu alloys with added Mn and Cr; however, AA7475 alloy only has a restricted amount of Mn added (0.06 wt% maximum) [5]. Y Zhou et al. conducted a study on the distribution of Mg and Cr in the dendrite arms of direct-chilled AA7475 ingots. Their findings indicate that the effective partition coefficients of Mg and Cr are 0.650 and 1.392, respectively, and illustrate the heterogeneous distribution of Mg and Cr along the radial direction of the dendrite arms’ cross-section [6]. Ahmed et al. have studied the homogenization treatment of Zr-containing AA7475 alloys. They give an optimum homogenization treatment so that dense and uniformly distributed Zr-bearing dispersoids can be obtained [7]. It is believed that the dispersoids particles are able to stabilize the grain structure and increase the recrystallization resistance by pinning grains and sub-grain boundaries [6,8]. The crack propagation resistance of Al-Zn-Mg-Cu alloys can be influenced by controlling grain structures. For instance, elongated pancake grains can effectively slow down crack growth when the crack is spreading at a right angle to the main axis of the grain. This results in transgranular propagation being preferred, which increases the energy dissipated during the damage process [9]. Tsai and Chuang studied the impact of grain size on the susceptibility to stress corrosion cracking (SCC) of the AA7475 alloy. Their findings indicated that homogeneous slip mode is a key factor in reducing the susceptivity of SCC. A finer grain size leads to a more homogeneous slip mode and smaller grain boundary precipitates, which makes the material more resistant to SCC [10]. The mechanism of hydrogen embrittlement in high-strength aluminum alloys containing various dispersoids was explored by M. Safyari et al. Their research demonstrates that the hydrogen level at the GBs/vacancies is considerably smaller in the Zr-added alloy than it is in the Cr-added alloy. They explained this as being caused by the elastic interaction between hydrogen and coherency stain [11].

The typical fracture toughness values for AA7475 alloy plate are around 40% higher when compared to those of AA7075 alloy in the same tempers [2]. According to a recent publication, it is possible to attain a fracture toughness (K_IC_) value of 65.2 MPa√m and a critical stress intensity factor (K_ISCC_) of 37.9 MPa√m for 7X75 alloy in the T73 temper [3]. It suggests that the crack propagation behavior can be controlled through the deliberate selection of alloy composition and thermal mechanical processing. AA7475 aluminum alloys are often available in T7351 and T7451 conditions. Its resistance to corrosion and ability to withstand fatigue are comparable to, and in some instances surpass, other high-strength aerospace alloys, such as AA7075, AA7050, and AA2024. This is also attributed to its lower Fe and Si content, as noted in various publications [4,12,13]. The Fe-containing phase in the Al-Zn-Mg-Cu series, such as Al_7_Cu_2_Fe or FeAl_m_, are brittle intermetallic particles that can nucleate cavities and cracks during the services period [14,15]. Verma et al. studied fatigue crack propagation (FCP) of AA7475 alloy. It was noted that the fatigue crack initiated from a surface grain, and the crack extension was dominated by ductile striations [16]. K. Wen and colleagues investigated the FCP behavior of an Al-Zn-Mg-Cu alloy with high zinc content under various aging conditions. They discovered that the FCP resistance improved during the aging process, which they attributed largely to changes in matrix precipitates [17].

Islam and Wallace have studied the retrogression and re-aging response of alloy AA7475. The results indicate that stress-corrosion-crack growth rates can vary independently of yield strength, and that T6 strength levels can be achieved in materials with stress-corrosion resistance comparable to that of the T73 condition for alloy AA7475 [18]. Their studies have also shown that stress-corrosion crack growth rates in retrogressed and re-aged materials are comparable to those in T73 temper [19]. Poole et al. claimed that the peak stress occurred after a short aging time. The acceleration of the kinetics of over-aging could be due to the deformation during aging [20]. Ohnishi et al. studied the retrogression and re-aging (RRA) process for the AA7475 alloy with the aim of enhancing its fracture toughness and SCC properties while preserving its high T6 level strength. Their findings indicated that the preservation of high strength was due to the forming of fine dispersions of age-hardening precipitates. The improvement in SCC resistance was attributed to the coarsening of grain boundary precipitates [4].

It should be noted that Cina and Ranis developed RRA in 1974 to improve corrosion resistance to that of the T73 temper while maintaining the peak aged strength [21]. The original study involved briefly heating 7075-T6 samples (from a few seconds to a few minutes) in the temperature range of 200–280 °C (reversion process). Today, the reversion temperatures have decreased to 160–180 °C, with the reversion time remaining in the range of tens of seconds or hundreds of seconds. For instance, T. Marlaud suggests a 20-min reversion process at 185 °C for their Al-Zn-Mg-Cu alloy study [22]. Nicolas discovered that the dissolution occurs rapidly and should be completed within 100 s if the retrogression process is carried out within a temperature range of 240–300 °C [23,24]. Neither scenario would be suitable for industrial production since the large plates require a significant amount of time to heat up or cool down evenly throughout their entire thickness. Additionally, the retrogression temperature must not be too low to avoid insufficient dissolution of precipitates.

We designed industrial-suited RRA experiment procedures to explore the impact of retrogression time on other mechanical properties, such as fracture toughness and fatigue limits. Comprehensive testing was conducted to examine the effect of RRA on these properties, with particular focus on the fatigue crack growth behavior of the alloy AA7475.

## 2. Experimental

The material is a modified AA7475 (namely 7X75) 80 mm hot-rolled plate produced by Guangxi Alnan Aluminium Inc. in China. The measured composition is Al-1.5Cu-2.6Mg-6.0Zn-0.2Cr-0.03Mn (wt%). The heat treatment procedures for the studied alloys are shown in Table 1.

The received plate was solution treated at a temperature of 473 °C for 4 h, then cooled rapidly in a Roller Hearth Furnace to minimize the transfer time. The plate was stretched immediately after quenching to minimize residual stress, then cut into 1000 mm × 1000 mm pieces for further aging processes. The T6 and T73 temper procedures are standard, without any modifications, and their results will serve as reference. However, the RRA process altered the retrogression time (which was changed from 0.5 to 5 h) while being performed at a temperature of 180 °C. The RRA process was conducted in a customer-designed aging furnace that has three isolated temperature zones, each set to designed temperatures. The furnace’s roller transports the aluminum plates from one temperature zone to another at a controlled speed, allowing for controlled soaking time. The temperature deviation in each zone is ±1 °C.

To fully understand the changes in properties, mechanical properties (ultimate tensile strength (UTS) and yield strength (YS)) tests were carried out along with the longitudinal direction. Dog-bone-shaped tensile specimens with a gauge length of 50 mm and diameter of 10 mm were used. Tensile tests were performed at a strain rate of 10^−3^ s^−1^ at room temperature. Five parallel tests were conducted.

A 60 mm thick compact tension (CT) (L-T direction) was used to measure the fracture toughness. The thickness ($B>2.5{\left(K_{IC}/\sigma_{YS}\right)}^{2}$) was selected to ensure that a valid plane-strain fracture toughness (K_IC_) was obtained. High-cycle fatigue (HCF) and Fatigue crack growth (FCG) tests were performed using a 110 kN MTS landmark hydraulic fatigue machine. Both L-T direction and T-L direction samples were tested. Hourglass-shaped specimens were utilized for high-cycle fatigue (HCF) testing, conforming to the size specified in ASTM standard E466-15. A 12.5-mm thick CT sample based on ASTM standard E647-15e1 was used for the FCG test with a load ratio of R = 0.06 at a maximum load of 7.0 kN and a frequency of 4.0 Hz. The crack length was measured using a Crack Opening Displacement (COD) gauge.

Microstructures were characterized using a ZEISS EVOMA10 scanning electron microscope (SEM) with an OXFORD Energy Dispersive Spectroscopy (EDS) and an Electron Backscattered Diffraction (EBSD) detector. The related EBSD data analysis was carried out using CHANNEL 5 software.

The FEI Tecnai G^2^ F20 Transmission electron microscope (TEM) equipped with an OXFORD INCA EDS detector was used to characterize the precipitation under different conditions. The TEM samples were circular discs 3 mm in diameter, cut from an 80 μm thick foil obtained through manual thinning. They were further polished using twin-jet electropolishing in a solution of 80% methanol and 20% nitric acid at a temperature below −25 °C. The particle sizes were measured through TEM image analysis using Image J software 1.53t. More than one hundred particles were analyzed for each condition.

## 3. Results and Discussions

### 3.1. Mechanical Properties, Fracture Toughness and Other Properties of Studied Alloy

The impact of retrogression time on strength, fatigue limits, and fracture toughness is depicted in Figure 1, with reference lines for T6 and T73 temper results. As seen in Figure 1a,b, both the UTS and YS decrease with increasing retrogression times, but remain within the range of T6 (565 MPa) and T73 (507 MPa) temper. It is also evident that the changes in YS are more pronounced than those in UTS at 2 h, 3 h, and 5 h, with a continuous decrease in YS and little change in UTS. The YS values for T6 and T73 temper are indicated by dashed lines in Figure 1b as 510 MPa and 410 MPa, respectively. Figure 1c displays a linear decrease in the fatigue limits with increasing retrogression time. It is evident that the fatigue limit (FL) exceeds the T6 temper (180 MPa) when retrogression occurs for 0.5 and 1 h. However, after 2 h of retrogression at 180 °C, the fatigue limit (FL) is equal to that of the T6 temper. While the fatigue limit decreases with retrogression time, it still exceeds the value for the T73 condition (170 MPa).

As depicted in Figure 1d, the fracture toughness shows a sharp rise with increasing retrogression time, particularly for retrogression times of 0.5 h, 1 h, and 2 h. Beyond a retrogression time of 2 h, the change in fracture toughness is more modest. The fracture toughness values fall within the range of T73 (64.7 MPa√m) and T6 temper (47.3 MPa√m) and the highest K_IC_ value is 63.5 MPa√m; this surpasses the traditional AA7475 report’s L-T value of 52 MPa√m and T-L value of 42 MPa√m [25]. Figure 1e displays exceptional K_ISCC_ values for samples subjected to RRA treatment; these are within the published range of 19–22 MPa√m s [26,27]. The K_ISCC_ value also increases with longer retrogression times, with a significant increase observed between 1 h and 2 h. Beyond that, the K_ISCC_ value stabilizes irrespective of retrogression time changes.

### 3.2. Fatigue Crack Growth Study

Figure 2 illustrates the FCGR curves (da/dN vs. ΔK) for alloys experiencing different retrogression times in the L-T direction (Figure 2a) and T-L direction (Figure 2b). It is shown that the crack growth rate along the long transverse direction (L-T) is less than the longitudinal direction (T-L). This can be seen by the crack growth value of ~0.08 × 10^−3^ mm/cycle (Figure 2a) in the L-T direction when compared to ~1 × 10^−3^ mm/cycle (Figure 2b) in the T-L direction at ΔK = 20 MPa√m. The result agrees with the data shown in other publications [16]. The decreased ΔK_th_ by about 2–3 units in the 2-h retrogression sample is also displayed in both figures. The da/dN value is more scattered when close to the ΔK_th_ value. This is because constant force amplitude test procedures are usually well suited for fatigue crack growth rates above 10^−6^ mm/cycle. The crack growth rate is, on average, smaller than on atomic spacing per cycle, and such a crack can be considered dormant. The COD method can barely record the crack change in this ΔK region. When a precise ΔK_th_ value is wanted, a K-decreasing test procedure is recommended in the standard [28]. We calculated the crack growth threshold by using the interpolation method. This involves taking the average of the five smallest da/dN values, as shown by the filled circles in Figure 2. The resulting mean ΔK value is estimated as ΔK_th_, which represents the fatigue threshold. The figures demonstrate that the ΔK_th_ value for RRA1h is higher than that for RRA2h in both directions, indicating that the alloy is more prone to cracking when subjected to longer retrogression treatment.

In the crack propagation state, which is characterized by higher ΔK values, the crack growth rate follows the Paris–Erdogan law [29].
(1)$$\frac{da}{dN}=C\cdot \Delta K^{m}$$

In this law, C and m are considered material constants that depend on factors such as frequency, temperature, and stress ratio. The fitted constants are displayed in the figures. The results show that the parameter C has a magnitude of approximately 1 × 10^−6^ for both L-T and T-L directions, except for the RRA2h sample in the T-L direction, where C has a value of 1 × 10^−7^. The coefficient m, which falls between 2.1–2.4, does not show significant variation. This m value is consistent with Verma’s findings for the same series of aluminum alloys [5]. It is interesting to compare the fatigue crack propagation (FCP) behavior with other alloys. Figure 2a includes the FCGR for alloy AA7050-T7451 from MMPDS for comparison [30]. The MMPDS data were obtained at a higher stress ratio (R = 0.1) and, therefore, the FCGR is expected to be higher than at R = 0.06 for the same ΔK values. However, as shown in the figure, the FCGR for both alloys is very similar in the low ΔK value range. In the high ΔK value range, the studied 7X75 alloy exhibits significantly lower values, indicating that it is highly resistant to fatigue crack growth in the L-T direction after retrogression treatment.

Figure 2b displays the fracture surface of the samples tested in T-L directions, roughly divided into three stages of crack behavior. However, the boundaries of each stage require clarification; these are the initial crack stage (I), the steady crack propagation stage (II), and the catastrophic failure stage (III). The initial crack stage and steady crack propagation stage appear brighter as their fracture surface is relatively smooth, while the catastrophic failure stage appears darker due to its unstable crack propagation. Further fractographic analysis is presented in the following section.

**Figure 2 materials-16-02733-f002:**
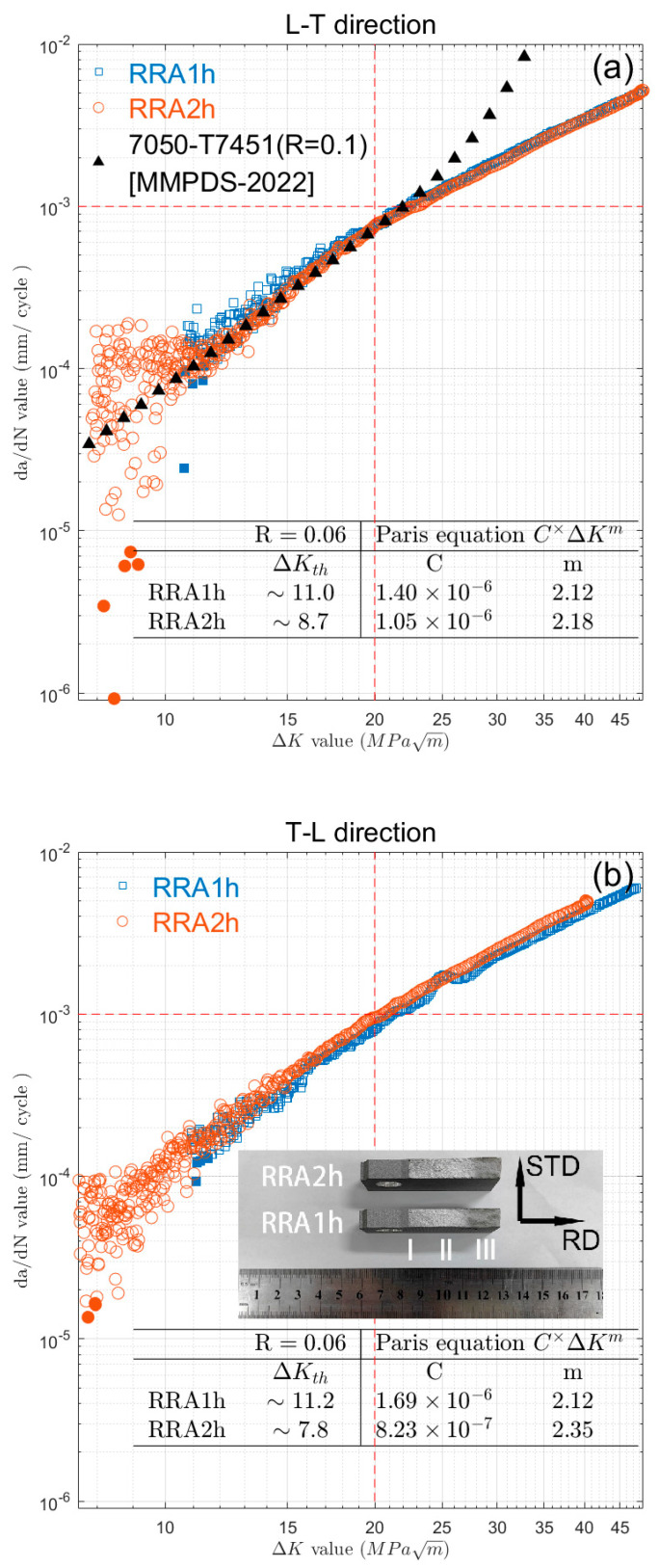
FCGR curves (da/dN vs. ΔK) for alloy retrogression at 180 °C for 1 h and 2 h for (**a**) L-T direction, the FCGR data of 7050-T7451 is from [30] and (**b**) T-L direction, the enclosed figure shows the fracture surface of tested samples.

### 3.3. Fractographic Characterizations

Figure 3 shows a detailed SEM characterization of the fractography of T-L samples at different retrogression times. The first column displays the fractography of RRA1h samples, and the second column shows the fractography of RRA2h samples.

The initial crack stage (Figure 3a,b) and steady crack propagation stage (Figure 3c,d) exhibit a typical cleavage fracture pattern, with straight cleavage cracks along the RD direction and some separations at certain angles. There is no evidence that the separations are linked to the crystallographic planes. Some prominent cracks exhibit the fiber grain structure, while the separations appear denser than in other conditions. RRA2h samples (Figure 3a) contain more micro-voids than RRA1h samples (Figure 3b); these result from plastic flow and will eventually enlarge during fracture. In stage II, the steady propagation of the main crack is less noticeable, and separated cracks at certain angles are the dominant behavior. Fracture voids are frequently observed in RRA2h conditions. The dimpled texture indicates better fracture toughness; this is consistent with the fracture toughness values in Figure 1.

Fatigue striations can be seen in both the crack initiation and propagation stages, with their morphology depicted in the top-right corner of images from Figure 3a–d. Striation spacings of approximately 430 nm were observed in the RRA1h condition, and approximately 315 nm in the RRA2h condition. Striations are traces left behind by the progressive expansion of a fatigue crack on the fracture surface. The location of the crack tip at the moment it was created is indicated by a striation. According to the observed striation spacings, FCGR is likely quicker in the RRA1h condition than the RRA2h condition. Thus, RRA1h samples should have lower fatigue limits than RRA2h samples. The fatigue limit, according to Figure 1c, is 180 MPa for the RRA2h condition and roughly 185 MPa for the RRA1h condition. In Figure 2b, the measured FCGR for the RRA1h condition is marginally lower than for the RRA2h condition. Hence, the striation spacing may thus necessitate a more thorough statistical analysis over the full fracture length. According to W.C. Connors, estimating the total number of striations would be more accurate if striation spacings were measured at a few different spots; though hundreds of striations are often present on fracture surfaces [31].

The formation of slip bands indicates a concentrated unidirectional slip on certain planes, causing a stress concentration. Typically, slip bands induce surface steps (e.g., roughness due persistent slip bands during fatigue) and a stress concentration which can be a crack nucleation site. Slip bands extend until impinged by a boundary, and the generated stress from dislocations pile-up against that boundary will either stop or transmit the operating slip depending on its (mis)orientation [32,33].

Figure 3e,f depict a catastrophic failure after the crack propagation stage. During this stage, fractography becomes more complicated. The stress intensity increases to the point where the crack grows rapidly, resulting in a complex fractography featuring both plastic and brittle fracture morphologies. Additionally, the fracture cavities are noticeably larger when compared to earlier stages.

### 3.4. Microstructure Characterizations and Discussions

The initial microstructure of a 1/4 thickness sample is displayed in Figure 4. It shows typical deformed grain structures with two planes that exhibit distinct grain structures. Figure 4a shows that the grains in the L-T plane have been squeezed due to rolling, with an aspect ratio (grain length in LTD/grain length in STD) of around 4.2. The T-L plane exhibits a pancake-like grain structure with a much higher aspect ratio of around 20.5. The EBSD image analysis reveals a low recrystallization fraction (less than 10%) due to the specific thermomechanical processing. This results in unrecrystallized grains that can dissipate crack growth energy and arrest crack paths [34]. The results in Figure 2 demonstrate that the FCGR is lower in the L-T direction compared to the T-L direction, leading to the conclusion that crack propagation is more favorable in pancake grains with high aspect ratio than in those with low aspect ratio. Many publications claim that the transgranular propagation along the pancake grains increases the energy dissipated during the damage process [9]. According to A. Garner, the high aspect ratio and large grain size produce little deflection or branching chances, which might lessen the local fracture tip stress intensity driving the crack propagation [35]. These findings have important ramifications for our knowledge of fracture propagation in Al-Zn-Mg-Cu alloys since the pancake grains were substantially larger and more complicated in shape and deformation structure in 3D when compared to what was previously believed from 2D.

The RRA process does not alter the grain structures of the alloys being examined. Many studies have demonstrated that retrogression leads to a decrease in strength initially, followed by a subsequent increase up to a secondary peak. This initial drop is typically the result of the dissolving of age-hardening precipitates [22,36,37]. However, the decrease is usually only temporary during the short duration of retrogression and does not impact industrial manufacturing processes.

The evolution of age-hardening precipitates during retrogression has been analyzed using TEM and is depicted in Figure 5. The first column shows the precipitates within the grains. The size of the precipitates does not vary significantly with increasing retrogression time, but the number density or volume fraction has significantly increased. The statistics of precipitates within grains are presented at the top of the figures. More than a hundred particles were analyzed for each condition. The average diameter increased from 6.13 nm at the RRA1h condition to 6.50 nm at the RRA5h condition. They are slightly larger than those at the T6 condition (approximately 4.8 nm) and smaller than those at the T7 condition (around 7.6 nm) [24]. The statistics also show that the number density of the precipitates increased with increasing retrogression time.

The second column in Figure 5 shows the precipitates along the grain/subgrain boundaries. It can be seen in Figure 5b,d,f that the width of the PFZs ranges from 30 to 50 nm and remains relatively unchanged with increasing retrogression time. The size of grain boundary precipitates (GBPs) is also within a similar range of 20 to 60 nm, similar to PFZs. The GBPs are widely dispersed rather than continuous along the grain/subgrain boundaries. This is considered beneficial for both SCC properties and fatigue propagation as dislocations can move to the surface, initiating cracks. Alternatively, dislocations can move to the PFZs, causing strain localization and stress concentrations during cyclic loading [25,38].

The crack propagation behavior is thought to be influenced by multiple factors, including the grain structure size, the size or number density of precipitates within grains, precipitates along grain boundaries, and the width of PFZs. Different crack propagation mechanisms may apply during different stages, and other strengthening mechanisms must also be taken into account. As a result, creating a sophisticated function that describes the relationship between microstructure parameters and fracture toughness or fatigue endurance limits is challenging. However, some models can help explain the fatigue crack growth behavior of aluminum alloys. For instance, K. Wen’s research primarily focused on the interaction between dislocations and age-hardening precipitates [6]. The study demonstrates that the rate of fatigue crack propagation (FCP) is proportional to the precipitate radius (*r*) and the reciprocal of the volume fraction (*fv*):(2)$$\frac{da}{dN}\propto r\cdot f_{v}^{-1/2}$$

The volume fraction of age-hardening precipitates is a crucial factor that affects the FCP and fracture toughness of the conditions studied. In this research, we propose an empirical relationship between fracture toughness and the volume fraction (*fv*) of age-hardening precipitates.
(3)$$f_{v}\approx \frac{C\cdot K_{IC}}{\sqrt{\sigma_{y}\cdot E\cdot d_{p}}}$$
where:

*C* is fitting constants; *K_IC_* is the fracture toughness in MPa√m; *σ_y_* is Yield strength; *E* is Young’s modulus; *d_p_* is the mean size of the particles; and *f_v_* is the volume fraction of the given type of precipitate. The calculated volume fraction of age-hardening precipitates is shown in Table 2. The equation provides good results, and the increasing trend agrees with the microstructure characterizations.

## 4. Conclusions

In the retrogression treatment process, the dissolution of fine and coherent precipitates that were previously formed during the initial aging treatment or natural aging occurs to some extent. The degree of dissolution is contingent on the temperature and duration of the retrogression treatment. This partial dissolution of precipitates leads to a more homogeneous distribution of solute atoms and vacancies, thereby increasing the driving force for the formation of new and smaller precipitates during subsequent aging. Consequently, the number density and volume fraction of η’ precipitates increase with increasing retrogression time, as supported by the findings presented in Table 2. As the size of age-hardening precipitates grows, their susceptibility to dislocation cutting decreases. This phenomenon, commonly known as the bypass mechanism, is well documented in literature [22,39,40]. It can result in the easy accumulation of dislocations in localized regions, and the FCGR for the RRA2h sample is correspondingly faster than that observed under other conditions.

In comparison to longer retrogression times, the size distribution of η’ precipitates is more varied when exposed to shorter retrogression times. As demonstrated by Figure 5a,c, both large and small precipitates can be observed. This dispersion of particles can result in a reduction of fatigue deformation, making it desirable to have a duplex precipitate structure in these types of alloys. Such a structure should consist of fine particles to provide high tensile properties and coarse particles to enhance fatigue strength.

Many other microstructures can influence FCGR. The presence of PFZs is well-known to be a soft region during cyclic stressing. While the η’ phase precipitates within the grains, the PFZs remain constant even with increasing retrogression time, as confirmed by the statistical analysis presented in Figure 5. The strength at grain boundaries and within individual grains therefore progressively balances, resulting in superior formability. These observations are consistent with the microstructural characterizations shown in Figure 3, where fracture voids are frequently observed in RRA2h conditions. Furthermore, the fracture toughness values presented in Figure 1 indicate that better fracture toughness can be achieved with prolonged retrogression treatments.

This paper studies the impact of retrogression time on the fatigue crack growth of a modified AA7475 aluminum alloy. To thoroughly examine the changes in properties with different RRA techniques, various tests, including tensile strength, fracture toughness, fatigue limits, etc., have been conducted. Results reveal that mechanical strength and fatigue limits decrease with increased retrogression time, while fracture toughness sharply increases, reaching its highest value of 63.5 MPa√m after a 5-h retrogression treatment. Prolonged retrogression increases the likelihood of crack initiation, and the crack growth rate is slower in the long transverse direction compared to the longitudinal direction due to the high aspect ratio of pancake-like grain structure in the T-L plane. An empirical relationship between fracture toughness and the volume fraction of age-hardening precipitates is proposed, and the number density of these precipitates increases with increased retrogression time as revealed by both the equation and microstructural characterization.

## Figures and Tables

**Figure 1 materials-16-02733-f001:**
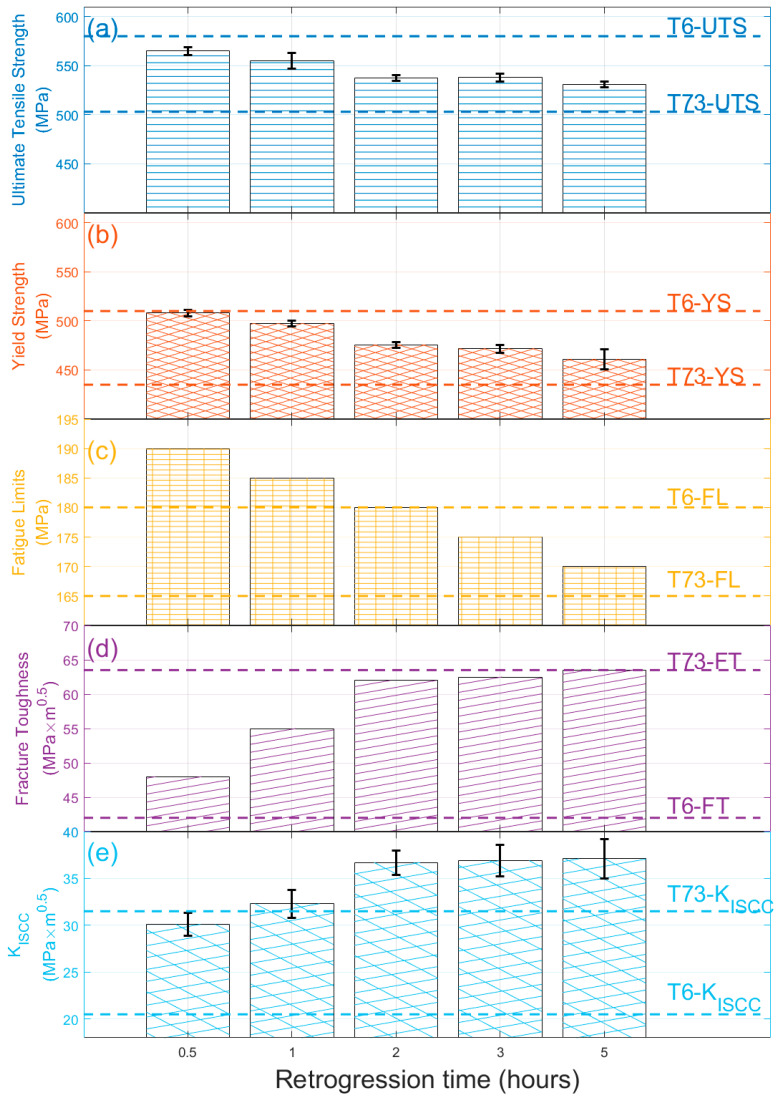
Effects of retrogression time at 180 °C on (**a**) ultimate tensile strength (UTS, MPa), (**b**) Yield strength (YS, MPa), (**c**) Fatigue endurance limits (FL, MPa), (**d**) Fracture toughness (FT, MPa√m), and (**e**) K_ISCC_ (MPa√m). Symbols “T6” and “T73” in the figure represented the benchmark of these conventional treatments.

**Figure 3 materials-16-02733-f003:**
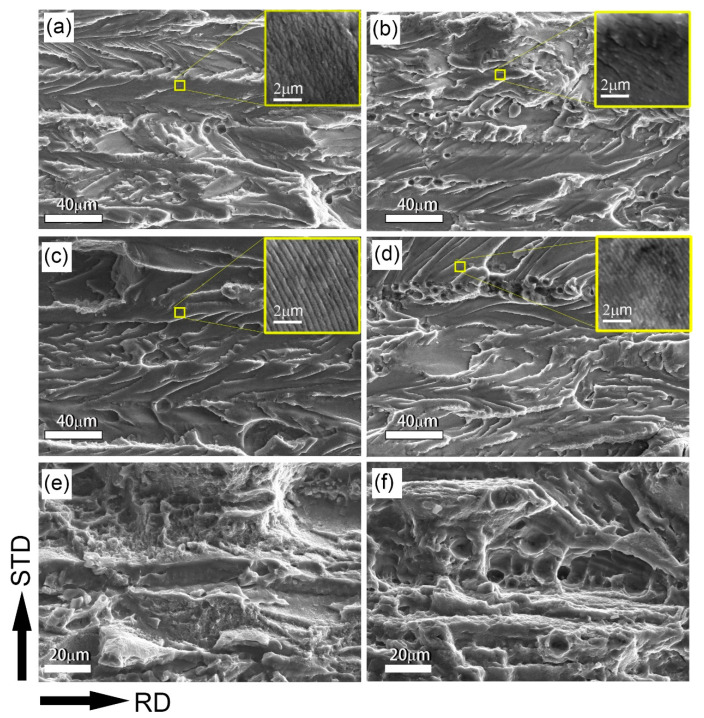
SEM characterization of fractography of T-L direction samples after retrogression at 180 °C for 1 h (first column) and 2 h (second column). Images (**a**,**b**) are taken from stage I-crack initial, (**c**,**d**) are taken from stage II-crack steadily propagation, and (**e**,**f**) are taken from stage III- catastrophic failure. The enclosed image in the top-right corner shows higher magnifications of the fatigue striations produced from amplitude loading.

**Figure 4 materials-16-02733-f004:**
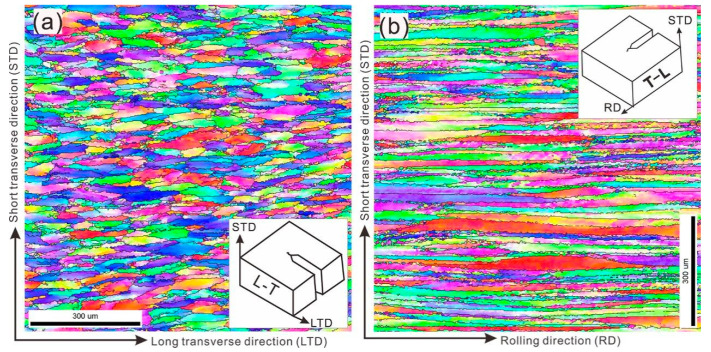
Inverse Pole Figures (IPF) show the grain structure of 7X75 aluminum alloy at 1/4 thicknesses of L-T direction (**a**) and T-L direction (**b**).

**Figure 5 materials-16-02733-f005:**
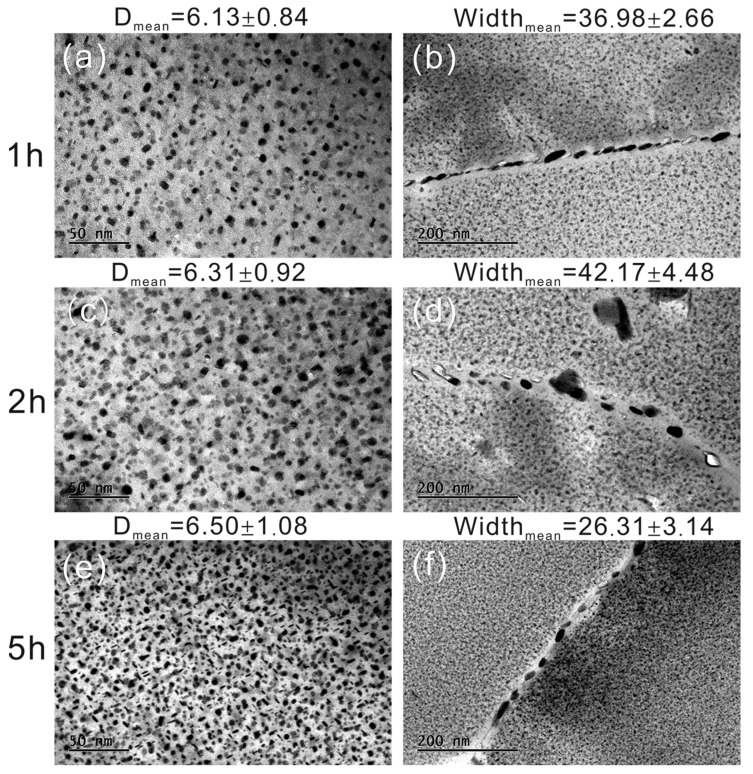
The age-hardening precipitates of RRA treated samples, (**a**,**b**) retrogression for 1 h, (**c**,**d**) retrogression for 2 h, and (**e**,**f**) retrogression for 5 h. The first column represents the precipitate within grains, the second column represents the precipitates on the grain/subgrain boundaries, and the third column shows the statistical analysis of the precipitates within grains.

**Table 1 materials-16-02733-t001:** Heat treatment procedures for studied materials.

Temper	Condition	Aging Process
T6	Peak-aged	24 h/121 °C
T73	Over-aged	6 h/121 °C + 30 h/163 °C
RRA	--	24 h/121 °C + (0.5–5) h/180 °C + 12 h/121 °C

**Table 2 materials-16-02733-t002:** The calculated volume fraction of η’ precipitates.

Retrogression Time (Hours)	Young’s Modulus (MPa)	Yield Strength (MPa)	K_IC_ (MPa√m)	Particle Size (nm)	Calculated Volume Fraction (%)
0.5	72,000	508	48	5.90	3.08
1	72,000	497.4	55	6.13	3.51
2	72,000	475.7	62	6.31	3.98
3	72,000	471.8	62.5	6.42	4.00
5	72,000	461	63.5	6.50	4.08

## Data Availability

We are unable to provide the data as it pertains to an ongoing project.

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
