# Peer review of "Influence of Retrogression Time on the Fatigue Crack Growth Behavior of a Modified AA7475 Aluminum Alloy"

_materials, 2023, doi:10.3390/ma16072733_

Round 1
Reviewer 1 Report
The manuscript in its current state could be accepted after major revision.
Before accepting the manuscript in its current state, some aspects should be corrected.
1.- In the table, specifically in the RRA temper line, the letter "h" is missing.
2.- List and describe the three formulas used in the manuscript.
3.- Figures 1 and 2 lack quality; they must be improved. For example, letters or numbers are spliced, and K1SCC should be changed to KISCC.
4.- Section 3.1 is a technical report; it lacks a discussion of the results obtained. Also, the elongation and electrical conductivity results are only mentioned in that section; going deeper into the results is necessary.
5.- The text that describes the figures must precede them.
6.- Improve the discussion of the following and compare it with the values obtained in the previous characterizations: "This may appear contradictory, as typically a larger striation spacing indicates a faster crack growth rate. However, it should be noted that the larger striation spacing in the RRA1h sample suggests a more rapid crack growth, leading to a quicker catastrophic failure. On the other hand, the smaller striation spacing in the RRA2h sample indicates steady crack propagation at a high ΔK value range. This is consistent with the image in Fig.3(b)."
7.-Although it is essential to determine a conclusion such as "crack propagation is more favorable in pancake grains with a high aspect ratio than in those with a low aspect ratio,o" it is necessary to add and discuss the reason for the results obtained; you need to explain this behavior.
8.-Discussion about precipitation behavior should be added.
Author Response
The manuscript in its current state could be accepted after major revision.
Before accepting the manuscript in its current state, some aspects should be corrected.
1.- In the table, specifically in the RRA temper line, the letter "h" is missing.
We have checked Table 1, the missing letter "h" has been added.
2.- List and describe the three formulas used in the manuscript.
The authors believe that we have provided an explanation of the formulas, including the Paris-Erdogan equation which can be found in line 229, while the remaining two equations can be found in lines 364-378.
3.- Figures 1 and 2 lack quality; they must be improved. For example, letters or numbers are spliced, and K1SCC should be changed to KISCC.
We have made some changes to Fig.1 and Fig.2 by restructuring the layout, changing the label styles, and increasing the image size for better quality. These modifications will enable readers to easily view the letters and numbers in the figures.
4.- Section 3.1 is a technical report; it lacks a discussion of the results obtained. Also, the elongation and electrical conductivity results are only mentioned in that section; going deeper into the results is necessary.
We concur that elongation and electrical conductivity have not been thoroughly discussed, as our emphasis is primarily on mechanical properties and fatigue crack behavior. We believe that the discussion should instead focus on providing detailed microstructural characterizations. We then included further discussions in subsequent sections.
5.- The text that describes the figures must precede them.
We concur with this proposal, but its success largely depends on the formatting of the publications.
6.- Improve the discussion of the following and compare it with the values obtained in the previous characterizations: "This may appear contradictory, as typically a larger striation spacing indicates a faster crack growth rate. However, it should be noted that the larger striation spacing in the RRA1h sample suggests a more rapid crack growth, leading to a quicker catastrophic failure. On the other hand, the smaller striation spacing in the RRA2h sample indicates steady crack propagation at a high ΔK value range. This is consistent with the image in Fig.3(b)."
We appreciate the feedback from the reviewers. We have acknowledged the inappropriate discussion and conducted a review of published articles. As a result, we have made revisions to the discussion section between lines 276-287.
7.-Although it is essential to determine a conclusion such as "crack propagation is more favorable in pancake grains with a high aspect ratio than in those with a low aspect ratio,o" it is necessary to add and discuss the reason for the results obtained; you need to explain this behavior.
We are grateful for the reviewer's input and have taken the time to examine some published works. Additionally, we have expanded our discussion on this topic in lines 318-325.
8.-Discussion about precipitation behavior should be added.
We have included further discussion regarding the precipitation behaviors between lines 381-393.

Reviewer 2 Report
The paper was well drafted and the data well explained, however, it is not suitable for publication in its' current form. The following are to be made:
Abstract
How the authors need to present quantitative results from their findings in the abstract.
Introduction
present the ref @line 100 appropriately.
Table description should come before the Table, see line 109-111.
Sentence hanging in line 126......"to fully understand the changes in property.?????????
Author Response
The paper was well drafted and the data well explained, however, it is not suitable for publication in its' current form. The following are to be made:
Abstract
How the authors need to present quantitative results from their findings in the abstract.
We have made changes to the abstract by including additional quantitative findings.
Introduction
present the ref @line 100 appropriately.
We observed that the citations were preceded by a full stop, but we have rectified the issue.
Table description should come before the Table, see line 109-111.
We have rearranged the sequence of the text and table.
Sentence hanging in line 126......"to fully understand the changes in property.?????????
We have rephrased this particular sentence.

Reviewer 3 Report
This manuscript is about "Influence of retrogression time on the fatigue crack growth behavior of a modified AA7475 aluminum alloy". The manuscript should be revised before publication based on the following comments.
1. In the introduction, it was reported that Zr and Cr can change the microstructure of the material. The following paper the effect of Zr and Cr in changing the mechanical properties and SCC (hydrogen embrittlement) behavior of 7xxx series aluminum alloys. This paper can be used in the introduction section.
https://doi.org/10.1016/j.corsci.2021.109895
2. It is recommended to add the KAM maps of the different specimens.
3. Further discussion should be added to explain the interaction between dislocations and precipitates of different sizes.
4. There are some slip bands in the fracture surfaces, the reason for that should be clarified.
Author Response
This manuscript is about "Influence of retrogression time on the fatigue crack growth behavior of a modified AA7475 aluminum alloy". The manuscript should be revised before publication based on the following comments.
1. In the introduction, it was reported that Zr and Cr can change the microstructure of the material. The following paper the effect of Zr and Cr in changing the mechanical properties and SCC (hydrogen embrittlement) behavior of 7xxx series aluminum alloys. This paper can be used in the introduction section.
https://doi.org/10.1016/j.corsci.2021.109895
We have gone through this article thoroughly and have referenced this valuable piece of research.
2. It is recommended to add the KAM maps of the different specimens.
KAM map of (a) RRA-1h-I, (b) RRA-2h-I, (c) RRA-1h-II, (d) RRA-2h-II, (e) RRA-1h-III, RRA-2h-III and statistical analysis results of KAM of the 6 investigated specimens
We conducted an EBSD analysis on alloys subjected to varying retrogression times, in response to the reviewer's suggestion. The accompanying images display the KAM maps of the crack propagation area, with the first column representing different crack stages for the RRA1h sample and the second column representing the RRA2h sample (I-crack initial stage, II-crack propagation stage, and III-catastrophic failure stage). Our findings suggest that retrogression time is inversely proportional to the density of geometrically-necessary dislocations (GNDs) and statistically-stored dislocations (SSDs). Furthermore, the slightly higher KAM values for the RRA1h sample suggest that it accumulates more plastic deformation during crack propagation, resulting in superior fatigue behavior compared to the RRA2h sample. However, this is a broad analysis of crack propagation behavior, and further characterization of microstructures such as dislocations or subgrains may be necessary to fully understand the differences in crack behavior. As a result, we plan to include KAM maps in our next article to provide a more comprehensive understanding of the underlying microstructural features.
Further discussion should be added to explain the interaction between dislocations and precipitates of different sizes.
We have included further discussion on precipitation behaviors between lines 381 and 393.
- There are some slip bands in the fracture surfaces, the reason for that should be clarified.
The feedback provided by the reviewers is greatly appreciated. We have conducted a thorough examination of published articles and have consequently expanded our discussion on slip bands, specifically in lines 293-298.

Round 2
Reviewer 1 Report
The manuscript in its current state could be accepted after major revision.
Before accepting the manuscript in its current state, some aspects should be corrected.
1.- The equations must be numbered within the manuscript.
2.- The manuscript's authors mention that the focus is providing detailed microstructure characterizations; hence the elongation and electrical conductivity results are not discussed. However, such results should be removed or justify inclusion through discussion.
3.- The text that describes the figures must precede them.
4.- The manuscript mentions, "According to the observed striation spacings, FCGR is likely quicker in the RRA1h condition than the RRA2h condition. Thus, RRA1h samples should have lower fatigue limits than RRA2h samples. The fatigue limit, according to Fig. 1(c), is 180 MPa for the RRA2h condition and roughly 185 MPa for the RRA1h condition. And in Fig.3 (b), the measured FCGR for the RRA1h condition is marginally lower than for the RRA2h condition. Hence, the striation spacing may thus necessitate a more thorough statistical analysis over the full fracture length. According to W.C. Connors, estimating the total number of striations would be more accurate if striation spacings were measured at a few different spots. Although hundreds of striations are often present on fracture surfaces [31]". The authors are correct and statistical analysis should be added to classify it as an acceptable result, just as they mentioned in the section on the precipitation analysis. The presentation of partial results decreases the quality of the work.
5.-The discussion on the behavior of the precipitation must be added. Specifically, it must be explained detailed what the increase in size and numerical density of the precipitates is attributed to, as well as the transformed fraction, if only results are presented without discussion on the manuscript falls into a technical report rather than a research paper.
Author Response
Before accepting the manuscript in its current state, some aspects should be corrected.
- - The equations must be numbered within the manuscript.
We have numbered every equations throughout the manuscript.
2.- The manuscript's authors mention that the focus is providing detailed microstructure characterizations; hence the elongation and electrical conductivity results are not discussed. However, such results should be removed or justify inclusion through discussion.
- We appreciate the suggestion from the reviewer, and as a result, we have removed the elongation and electrical conductivity results and their corresponding descriptions. Consequently, we have deleted Figure 2 and have included the KISCC results in Figure 1.
- - The text that describes the figures must precede them.
We have rearranged the order of figures and their corresponding descriptions.
- - The manuscript mentions, "According to the observed striation spacings, FCGR is likely quicker in the RRA1h condition than the RRA2h condition. Thus, RRA1h samples should have lower fatigue limits than RRA2h samples. The fatigue limit, according to Fig. 1(c), is 180 MPa for the RRA2h condition and roughly 185 MPa for the RRA1h condition. And in Fig.3 (b), the measured FCGR for the RRA1h condition is marginally lower than for the RRA2h condition. Hence, the striation spacing may thus necessitate a more thorough statistical analysis over the full fracture length. According to W.C. Connors, estimating the total number of striations would be more accurate if striation spacings were measured at a few different spots. Although hundreds of striations are often present on fracture surfaces [31]". The authors are correct and statistical analysis should be added to classify it as an acceptable result, just as they mentioned in the section on the precipitation analysis. The presentation of partial results decreases the quality of the work.
After reviewing multiple published articles, we found that there is no clear consensus regarding the relationship between striation spacings and the fatigue crack growth rate (FCGR). Some studies suggest that there may be a linear relationship between the two, while others propose more complex relationships. For example, Wanhill (Metallurgical Transactions A volume 6, 1587, 1975) claimed that only fatigue in air can possible be related to dislocation substructure striation spacings below the fracture surface. Meanwhile, Saanouni et al. (Materialia, 101753, 2023) suggest that striation spacing is also associated with recrystallized grains. Some studies have found that the striation spacing decreases with increasing crack growth rate, as faster-growing cracks tend to leave more closely spaced striations than slower-growing cracks. Given this complexity, we chose to focus solely on characterizing the microstructure of our samples and refrained from making any definitive conclusions regarding the relationship between striation spacings and FCGR. We believe that this topic warrants further investigation and may be a topic for future research.
- -The discussion on the behavior of the precipitation must be added. Specifically, it must be explained detailed what the increase in size and numerical density of the precipitates is attributed to, as well as the transformed fraction, if only results are presented without discussion on the manuscript falls into a technical report rather than a research paper.
We have revised the discussion section between lines 370-400. We have covered the changes that occur in age-hardening precipitates during retrogression, the FCGR in different conditions, and the impact of PFZs.
Reviewer 3 Report
The manuscript can be published.
Author Response
We have made a few English changes.